# Respiratory Arousals in Patients with Very Severe Obstructive Sleep Apnea and How They Change after a Non-Framework Surgery

**DOI:** 10.3390/healthcare10050902

**Published:** 2022-05-13

**Authors:** Ethan I. Huang, Shu-Yi Huang, Yu-Ching Lin, Chieh-Mo Lin, Chin-Kuo Lin, Chia-Yu Hsu, Ying-Chih Huang, Jian-An Su

**Affiliations:** 1Department of Otolaryngology, Chang Gung Memorial Hospital, Chiayi 61363, Taiwan; 2Sleep Center, Chang Gung Memorial Hospital, Chiayi 61363, Taiwan; 8802022@cgmh.org.tw (S.-Y.H.); lin0927@cgmh.org.tw (Y.-C.L.); 3School of Medicine, Chang Gung University, Taoyuan 33302, Taiwan; 4Division of Pulmonary and Critical Care Medicine, Chang Gung Memorial Hospital, Chiayi 61363, Taiwan; f124510714@cgmh.org.tw (C.-M.L.); lingh@cgmh.org.tw (C.-K.L.); 5Department of Nursing, Chang Gung University of Science and Technology, Chiayi 61363, Taiwan; 6Department of Respiratory Care, Chang Gung University of Science and Technology, Chiayi 61363, Taiwan; 7Graduate Institute of Clinical Medical Sciences, College of Medicine, Chang Gung University, Taoyuan 33302, Taiwan; 8Department of Neurology, Chang Gung Memorial Hospital, Chiayi 61363, Taiwan; mr8898@cgmh.org.tw (C.-Y.H.); ngingchi@cgmh.org.tw (Y.-C.H.); 9Department of Psychiatry, Chang Gung Memorial Hospital, Chiayi 61363, Taiwan; sujian@cgmh.org.tw

**Keywords:** palatoplasty, one-stage, retropharynx, polysomnography, Continuous-Positive-Airway-Pressure (CPAP)

## Abstract

Respiratory arousal is the change from a state of sleep to a state of wakefulness following an apnea or hypopnea. In patients with obstructive sleep apnea (OSA), it could have a helpful role to activate upper airway muscles and the resumption of airflow and an opposing role to contribute to greater ventilatory instability, continue cycling, and likely exacerbate OSA. Patients with very severe OSA (apnea-hypopnea index (AHI) ≥ 60 events/h) may have specific chemical (e.g., possible awake hypercapnic hypoxemia) and mechanical (e.g., restricted dilator muscles) stimuli to initiate a respiratory arousal. Little was reported about how respiratory arousal presents in this distinct subgroup, how it relates to AHI, Epworth Sleepiness Scale (ESS), body mass index (BMI), and oxygen saturation, and how a non-framework surgery may change it. Here, in 27 patients with very severe OSA, we show respiratory arousal index was correlated with each of AHI, mean oxyhemoglobin saturation of pulse oximetry (SpO2), mean desaturation, and desaturation index, but not in BMI or ESS. The mean (53.5 events/h) was higher than other reports with less severe OSAs in the literature. The respiratory arousal index can be reduced by about half (45.3%) after a non-framework multilevel surgery in these patients.

## 1. Introduction

Arousal is the change from a state of sleep to a state of wakefulness [1]. In patients with OSA, respiratory arousal is one following an obstructive apnea (airway occlusion) or hypopnea (airway narrowing) of sufficient length or severity to result in hypoventilation. Respiratory arousal could have two opposing roles, a helpful role vs. a harmful one. It may be the key to terminate an apnea. Viewed in this way, arousal from sleep is a lifesaving event. Unfortunately, if arousals are too frequent, sleep is disrupted and not restorative, even if the total sleep time is near normal [2].

Respiratory arousal viewed as a helpful event is believed to be critical to activate upper airway muscles and the resumption of airflow [3,4,5]. As Phillipson reported, the ability to arouse may be the most important response to a respiratory stimulus, a crucial and potentially life-saving response [1]. A respiratory stimulus can be chemical (hypercapnic hypoxemia) [6,7,8,9,10], neurogenic, or mechanical [11,12,13,14], developing during periods of obstructive apnea or hypopnea [6]. Respiratory arousal may enable gradual dilator muscle activation and improvements in airflow such that stable breathing can be attained [15]. Although studies showed arousal is not required for an adequate flow response, flow response was higher when arousals occurred (e.g., see Younes [11]).

In contrast, repeated respiratory arousals may be harmful in several ways. They promote an unnecessarily high flow response at upper airway opening, contribute to the progress of greater ventilatory instability [11], help to continue cycling and likely exacerbate OSA [11,16,17,18], and disrupt sleep by producing prolonged awakenings and thus shortening total sleep time [6]. However, repeated short arousals can increase daytime sleepiness without considerably reducing total sleep time [19,20,21]. The sleep disruption resulting from repeated arousals plays a major role in the pathogenesis of most of the consequences of OSA (i.e., neuropsychiatric. respiratory, and cardiovascular) and may contribute to the progression of OSA severity [22]. For example, arousal index serves as a marker of carotid artery atherosclerosis in OSA patients [23]. Preventing these arousal-associated consequences is one of the main goals of sleep surgeries, especially in patients with very severe OSA as respiratory arousals are associated with OSA severity [11,22].

Patients with very severe OSA generally have a disadvantaged anatomy, a confined retroglossal space and framework [24] with a smaller cross-sectional area [25]. Their lateral pharyngeal walls move comparatively little in inspiration [25]. It is unclear how this anatomy affects the mechanical stimulus on respiratory arousals. Their daytime awake partial pressure of oxygen may go as low as 77 mmHg [26]. Little is known about this specific hypercapnic hypoxemia and how this chemical stimulus influences respiratory arousals. It is unclear how respiratory arousals induced by the distinct mechanical or chemical stimuli in this subgroup associate with the patients’ Epworth Sleepiness Scale (ESS), body mass index (BMI), or oxygen saturation. Furthermore, their narrow airway is difficult to enlarge via conventional uvulopalatopharyngoplasty (UPPP) or non-framework surgeries [27,28]. A non-framework surgery is one that does not involve skeletal bones. It can reduce or re-shape the soft tissues from the nasal cavity, adenoid, palate and oropharynx, and tongue, but is limited to preserving the functions of soft tissues managed. As studies reported various effects of surgery on arousal [28,29,30,31,32], including a non-significant change of arousal index after UPPP in patients with moderate OSA [31], it is uncertain what may happen on respiratory arousal after a non-framework surgery in patients with very severe OSA.

Little was reported about how respiratory arousal presents in the distinct subgroup of very severe OSA, how it relates to AHI, ESS, BMI, and oxygen saturation, and how a non-framework surgery may change it. In patients with AHI ≥ 60 events/h, we tested the correlations of respiratory arousal vs. each of the sleep parameters mentioned above. We also compared respiratory arousal index before and after a non-framework multilevel surgery in these patients with disadvantaged anatomy.

## 2. Materials and Methods

### Ethical Statements

The Institutional Review Board (IRB) of Chang Gung Medical Foundation, Taiwan approved the study methods and protocols (IRB number: 202001198B0). We performed the study under Good Clinical Practice and the laws and regulations. As a retrospective cohort study, the IRB approved the waiver of the participants’ consent.

We investigated the same set of patients enrolled between March 2015 and January 2018 as in our earlier works on very severe OSA [33,34]. Participants included those who met the criteria of 1. age 20 or more, 2. AHI 60 events/h or more, 3. bad compliance of CPAP or high on-CPAP AHI referred from a sleep medicine specialist, 4. received a one-stage multi-level sleep surgery with the modified Z-palatoplasty performed with one-layer closure and open partial tongue-base glossectomy [35], and 5. available preoperative and postoperative polysomnographies (PSGs) for required recordings.

Every patient completed a PSG 6 months (193 ± 67 days) after the surgery, when the postoperative anatomy was considered stable (as the period of 3 to 6 months was adopted in the past reports, e.g., see Li, H. Y. et al. [36] and Hessel, N. S. & de Vries, N. [37]). Those who did not complete a postoperative PSG were excluded. Each PSG was conducted overnight in the level-1 sleep laboratory of the authors’ tertiary referral hospital by a licensed technician and interpreted by a sleep specialist. Respiratory arousal index is the number of average hourly cortical arousals following an apnea or hypopnea. An apnea is defined as the complete cessation of airflow for at least 10 s [33]. A hypopnea is defined as a decrease in airflow ≥ 30% for at least 10 s that is accompanied by either Electroencephalography (EEG) signs of arousal or by a 4% or greater decrease in oxygen saturation [33].

We plotted a scatter graph to exclude outliers and justify the application of a correlation coefficient, then calculated the correlation coefficient to show the relationship between respiratory arousal index and each of AHI, ESS, BMI, mean SpO2, mean desaturation, and desaturation index. A regression line was attached when there was statistical significance. We performed a paired t-test to examine the change of respiratory arousal index, against no change after the surgery, and did the same on total arousal index (that includes spontaneous arousals and respiratory effort–related arousal (RERA)) to compare the results with those reported in the literature. The statistical significance was tested as α = 0.05.

The statistical examinations were performed in MATLAB 9.4.0.813654 (MathWorks, Natick, MA, USA).

## 3. Results

There were 27 patients (24 men and 3 women) aged 29 to 63 years (mean = 47.4 ± a standard deviation (SD) of 10.6). Their respiratory arousal indexes ranged from 16.4 to 85.5 events/h (mean 53.5 ± 16.0) and AHIs ranged from 60.8 to 94.4 events/h (mean 73.8 ± 10.3). Figure 1 shows the scatter plot and correlation of respiratory arousal index vs. AHI. The correlation for the data showed that respiratory arousal index was positively related to AHI in these very severe OSA patients, r = 0.543, *p* = 0.003, two-tailed test. The coefficient of determination = 0.295, which indicates that 29.5% of the variance in AHI can be explained by respiratory arousal index.

Their ESS ranged from 1 to 18 (mean 9.6 ± 5.2). Figure 2 shows the scatter plot and correlation of respiratory arousal index vs. ESS. The correlation for the data showed that respiratory arousal index was not related to ESS in these very severe OSA patients, r = 0.106, *p* = 0.597, two-tailed test.

Their BMIs ranged from 21.8 to 37.6 kg/m^2^ (mean 28.5 ± 3.5). Figure 3 shows the scatter plot and correlation of respiratory arousal index vs. BMI. The correlation for the data showed that respiratory arousal index was not related to BMI in these very severe OSA patients, r = 0.361, *p* = 0.065, two-tailed test.

Their mean SpO2 ranged from 82.6 to 96.5 events/h (mean 92.3 ± 3.4). Figure 4 shows the scatter plot and correlation of respiratory arousal index vs. mean SpO2. The correlation for the data showed that respiratory arousal index was negatively related to mean SpO2 in these very severe OSA patients, r = −0.394, *p* = 0.042, two-tailed test. The coefficient of determination = 0.155, which indicates that 15.5% of the variance in mean SpO2 can be explained by respiratory arousal index.

Their mean desaturation ranged from 4.1 to 22 % (mean 10.0 ± 5.0). Figure 5 shows the scatter plot and correlation of respiratory arousal index vs. mean desaturation. The correlation for the data showed that respiratory arousal index was positively related to mean desaturation in these very severe OSA patients, r = 0.526, *p* = 0.005, two-tailed test. The coefficient of determination = 0.277, which indicates that 27.7% of the variance in mean desaturation, can be explained by respiratory arousal index.

Their desaturation index ranged from 34.1 to 85.3 events/h (mean 62.5 ± 12.5). Figure 6 shows the scatter plot and correlation of respiratory arousal index vs. desaturation index. The correlation for the data showed that respiratory arousal index was positively related to desaturation index in these very severe OSA patients, r = 0.552, *p* = 0.003, two-tailed test. The coefficient of determination = 0.305, which indicates that 30.5% of the variance in mean desaturation index, can be explained by respiratory arousal index.

All patients underwent Z-palatoplasty and partial open tongue-base glossectomy. One patient had UPPP at another hospital. Twenty-five received septomeatoplasty. One and three patients underwent regular adenoidectomy and endoscopic sinosurgery, respectively. See our earlier works for details [33,34,35]. Figure 7a illustrates the individual respiratory arousal index reductions and a five-number summary of pre- and postoperative respiratory arousal indexes. The surgery reduced respiratory arousal index from 53.5 ± 16.0 to 23.5 ± 18.3. The paired t-test showed *p* < 0.001. Figure 7b illustrates those on total arousal indexes. The surgery reduced total arousal indexes from 57.0 ± 14.6 to 31.2 ± 17.0, *p* < 0.001.

## 4. Discussion

In these patients with very severe OSA (mean AHI 73.8 ± 10.3 events/h) having their specific chemical (hypercapnic hypoxemia pattern) and mechanical (restricted dilator muscles) stimuli, the mean respiratory arousal index was 53.5 with a SD of 16.0 events/h. The mean respiratory arousal index was higher than Goh’s 47.1 events/h (with a mean AHI of 70.7 events/h) [29], Verse’s 36.3 events/h (with a mean AHI of 38.9 events/h), and Boudewyns’ 24.6 events/h (with a mean respiratory disturbance index (RDI) of 19.7 events/h). Even within the subgroup of very severe OSA, the results show respiratory arousal index was positively related to AHI (Figure 1).

In addition to AHI, the results also show correlation of respiratory arousal index vs. each of mean SpO2, mean desaturation, and desaturation index, but not in BMI or ESS. In these patients, the respiratory arousal index explained 29.5%, 15.5%, 27.7%, and 30.5% of the variance in AHI, mean SpO2, mean desaturation, and mean desaturation index, respectively. The results of correlation are compatible with reports [38,39,40] in the literature, except for BMI. Although our sample size of 27 is larger than Goh’s 11 patients [29] and Boudewyns’ 10 patients [31], it is small and limits the generalizability of the results (e.g., on correlation with BMI). The result of the respiratory arousal index not being associated with BMI and ESS suggests that it is less associated with obesity and drowsiness. It requires future studies with a large sample size and studies investigating inspiratory effort at end-apnea in patients with very severe OSA to verify the observations.

Another concern for this study is arousal intensity. Cortical arousals from sleep are quantified as an all or none phenomenon by the American Academy of Sleep Medicine scoring rules [41]. Yet not all cortical arousals are identical. The factors affecting arousal intensity and its potential role in sleep apnea pathogenesis were unclear. Arousal intensity is a distinct trait and an important mediator of ventilatory and pharyngeal dilator muscle responses to arousal [42]. The respiratory arousals before and after the surgery may change when arousal intensity is considered.

Surgery as a treatment for sleep fragmentation and its consequences in OSA is to minimize the occurrence of the arousal-provoking stimuli that produce sleep disruption, by maintaining upper airway patency and normalizing blood gases and inspiratory effort during sleep [22]. The non-framework multilevel surgery in this study reduced total arousal index from 57.0 to 31.2 events/h. The reduction rate of 45.3% is better than Boudewyns’ 1.6% after UPPP [31], but not as good as Goh’s 67.9% after Maxillomandibular Advancement (MMA) [29] and Verse’s 51.8% after a multilevel surgery of uvula flap, tonsillectomy, hyoid suspension, and radiofrequency treatment of the tongue base in patients with moderate to severe OSA [30]. It is difficult to compare the results with the reports in the literature because of the lack of controls, such as disease severity and surgical procedures. The common ground is that a sleep surgery usually reduces the frequency of arousals.

The level of how CPAP therapy reduces arousal index varies [43,44,45]. In Kim’s study [45], arousal index was reduced 87.6% from 57.1 to 7.1 events/h in 34 patients with severe OSA (mean 63.4 ± 17.5 event/h). Ferguson et al. reported a reduction rate of 23.9% from 28.9 to 22.0 events/h in 27 patients with moderate to severe OSA (mean AHI 24.5 ± 8.8 events/h) [43]. Randerath et al. reported a reduction rate of 35.3% from 21.8 to 14.1 events/h in 20 patients with mild to moderate OSA (mean AHI 17. 5 ± 7.7 events/h) after using CPAP for 6 weeks [44]. It is difficult to compare these CPAP results with the surgery in the present study because of the lack of controls, such as disease severity, patient selection, and home or sleep center monitoring. For patients with an acceptable compliance with CPAP but ask for the surgery, a future study is required to investigate how CPAP therapy may improve arousal in these patients before the surgery. The results may help the decision-making in the preoperative discussion.

## 5. Conclusions

In patients with very severe OSA having their specific chemical (e.g., possible awake hypercapnic hypoxemia) and mechanical (e.g., restricted dilator muscles) stimuli, the mean respiratory arousal index may be higher than other reports with less severe OSAs. The results show that respiratory arousal index was correlated with AHI, mean SpO2, mean desaturation, and desaturation index, but not in BMI or ESS in the data of this study. The respiratory arousal index can be reduced by about half (45.3%) after a non-framework multilevel surgery in these patients with disadvantaged anatomy.

## Figures and Tables

**Figure 1 healthcare-10-00902-f001:**
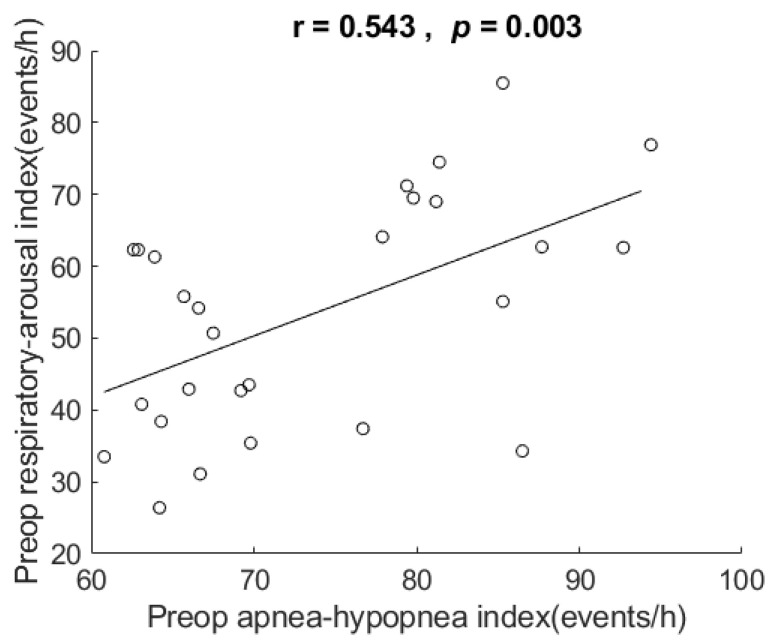
Scatter plot and correlation of respiratory arousal index vs. apnea-hypopnea index (AHI). It shows respiratory arousal index was positively related to AHI in the very severe OSA patients in this study. Preop: preoperative. h: hour.

**Figure 2 healthcare-10-00902-f002:**
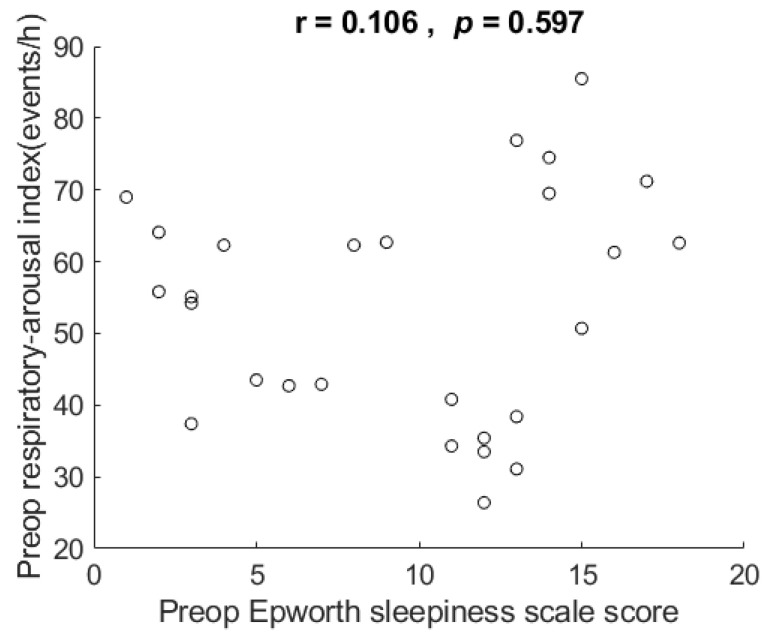
Scatter plot and correlation of respiratory arousal index vs. Epworth Sleepiness Scale (ESS). It shows respiratory arousal index was not related to ESS in the very severe OSA patients in this study. Preop: preoperative. h: hour.

**Figure 3 healthcare-10-00902-f003:**
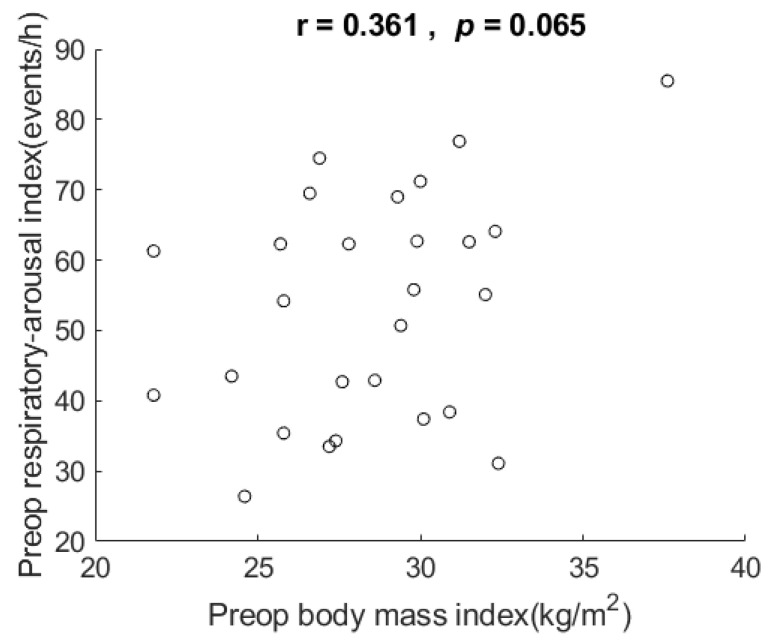
Scatter plot and correlation of respiratory arousal index vs. body mass index (BMI). It shows respiratory arousal index was not related to BMI in the very severe OSA patients in this study. Preop: preoperative. h: hour.

**Figure 4 healthcare-10-00902-f004:**
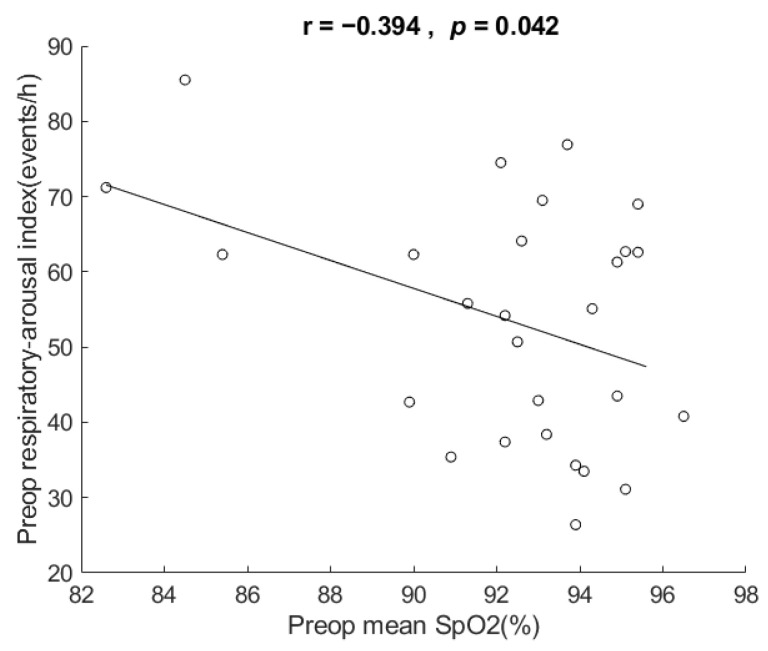
Scatter plot and correlation of respiratory arousal index vs. mean oxyhemoglobin saturation of pulse oximetry (SpO2). It shows respiratory arousal index was negatively related to mean SpO2 in the very severe OSA patients in this study. Preop: preoperative. h: hour.

**Figure 5 healthcare-10-00902-f005:**
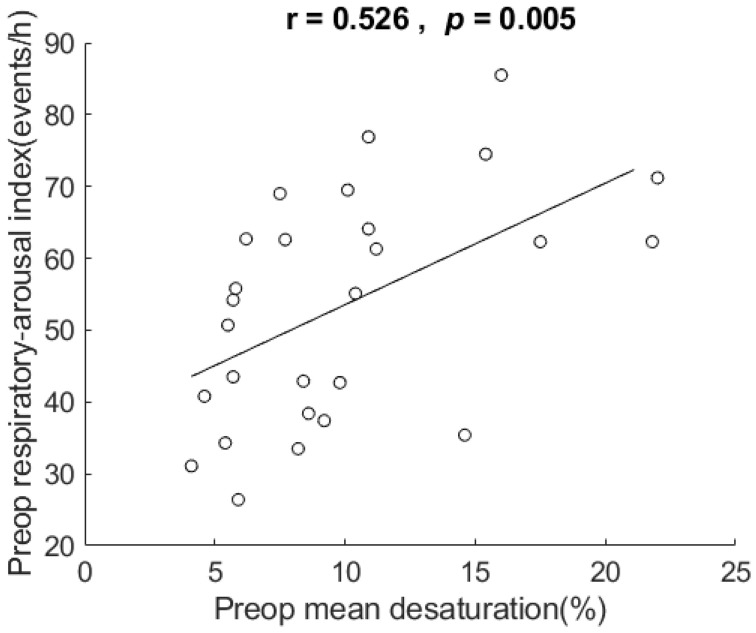
Scatter plot and correlation of respiratory arousal index vs. mean desaturation. It shows respiratory arousal index was positively related to mean desaturation in the very severe OSA patients in this study. Preop: preoperative. h: hour.

**Figure 6 healthcare-10-00902-f006:**
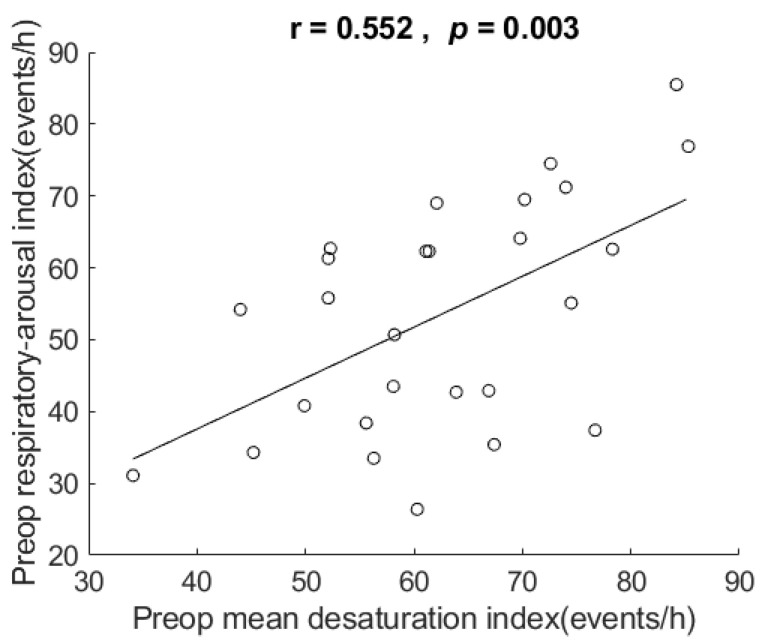
Scatter plot and correlation of respiratory arousal index vs. desaturation index. It shows respiratory arousal index was positively related to desaturation index the very severe OSA patients in this study. Preop: preoperative. h: hour.

**Figure 7 healthcare-10-00902-f007:**
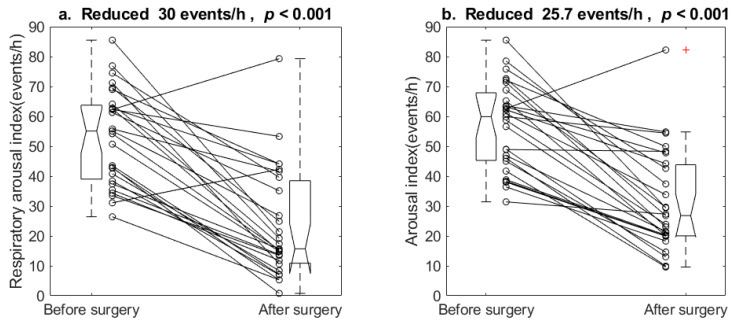
Individual change of (**a**) respiratory arousal index and (**b**) total arousal index before and after the surgery. The surgery reduced both respiratory and total arousal indexes. h: hour.

## Data Availability

The data presented in this study are available on request from the corresponding author. The data are not publicly available due to privacy.

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
