# Peer review of "Respiratory Arousals in Patients with Very Severe Obstructive Sleep Apnea and How They Change after a Non-Framework Surgery"

_healthcare, 2022, doi:10.3390/healthcare10050902_

Round 1

Reviewer 1 Report

Introduction 

  1. I feel the authors should add more about non-framework surgeries the merit and demerit, keeping in mind the word limit for the introduction
  2. Reconstruct the objectives, the need of the study needs more clarity 

Methods 

  1. the ethical clearance and informed consent should start first
  2. Why are authors only looking for the association? 
  3. why Not authors also check for regression to know how much the independent factor influences the dependent factor and by how much 
  4. I feel authors should include a section on data analysis where they can  clearly define what stats or software how was used 
  5. Exclusion criteria need more clarity 
  6. Abbreviate PSG in text 
  7. It will be good if authors can tabulate a data on baseline measurements like age, duration of the problem
  8. Why the authors just do association why can't PrePost changes be seen with RANOVA or so 

Discussion 

1. Discussion needs to be modified accordingly 

Author Response

Enclosed is our revised manuscript. We have addressed the questions. The response to each question is listed as follows. Please note the reference numbers of the responses are indexed at the end of this letter and differ from those in the manuscript.

Introduction 

  1. I feel the authors should add more about non-framework surgeries the merit and demerit, keeping in mind the word limit for the introduction

We added in the paragraph of Introduction: “… A non-framework surgery is one that does not involve skeletal bones. It can reduce or re-shape the soft tissues from the nasal cavity, adenoid, palate and oropharynx, and tongue, but is limited to preserve the functions of soft tissues managed. As studies reported various effects of surgery on arousal...”

  1. Reconstruct the objectives, the need of the study needs more clarity 

We rewrote the last paragraph of Introduction to: “Little was reported about how respiratory arousal presents in the distinct subgroup of very severe OSA, how it relates to AHI, ESS, BMI, and oxygen saturation, and how a non-framework surgery may change it. Here, in patients with AHI ≥ 60 events/hour, we tested the correlations of respiratory arousal vs. each of the sleep parameters mentioned above. We also compared respiratory arousal index before and after a non-framework multilevel surgery in these patients with disadvantaged anatomy.”

Methods 

  1. the ethical clearance and informed consent should start first

We moved the statement to the first paragraph.

  1. Why are authors only looking for the association? 

The aim of this study is to report how respiratory arousal presents in the distinct subgroup of very severe OSA, how it relates to AHI, ESS, BMI, and oxygen saturation, and how a non-framework surgery may change it. A scatter plot is a good way to show not only the relationship but also individual data. Each of the scatter plot we reported illustrates individual points of respiratory arousal index and one of the above sleep parameters, with the correlation between these 2 variables. Please let us know specifically if there is a way to improve.

  1. why Not authors also check for regression to know how much the independent factor influences the dependent factor and by how much 

To show how one sleep parameter is explained by respiratory arousal index, we added the result of coefficient of determination for each figure with a p < 0.05 in Results. In Figure 1, “The coefficient of determination = 0.295, which indicates that 29.5% of the variance in AHI can be explained by respiratory arousal index.” In Figure 4, “The coefficient of determination = 0.155, which indicates that 15.5% of the variance in AHI can be explained by respiratory arousal index”. In Figure 5, “The coefficient of determination = 0.277, which indicates that 27.7% of the variance in mean desaturation can be explained by respiratory arousal index”. In Figure 6, “The coefficient of determination = 0.305, which indicates that 30.5% of the variance in mean desaturation index can be explained by respiratory arousal index”.

The regression lines are as follows. In Figure 1: Y = -8.97+0.85X; In Figure 4: Y=224.95-1.86X; In Figure 5: Y=36.61+1.69X; In Figure 6: Y=9.23+0.71X. We did not add the information to prevent distractions. We will add them in if the journal decides to do so.

  1. I feel authors should include a section on data analysis where they can  clearly define what stats or software how was used 

As included in Methods. We calculated the correlation coefficient to show the relationship between respiratory arousal index and each of AHI, ESS, BMI, mean SpO2, mean desaturation, and desaturation index. A regression line was attached when there was a statistical significance. We performed a paired t-test to examine the change of respiratory arousal index, against no change after the surgery The statistical significance was tested as α = 0.05. The statistical examinations were performed in MATLAB 9.4.0.813654 (MathWorks, Natick, Massachusetts, U.S.A.). Please indicate specifically if there are any stats or data analysis can be better stated.

  1. Exclusion criteria need more clarity 

As the 5th inclusion criterium is “available preoperative and postoperative PSGs for required recordings”, those did not complete a postoperative PSG were excluded. We added this exclusion statement in the Methods.

  1. Abbreviate PSG in text 

We moved the abbreviation from Keywords to the text.

  1. It will be good if authors can tabulate a data on baseline measurements like age, duration of the problem

We reported the mean and range of age in the 1st paragraph of Result. We do not have accurate data about the duration of the problem.

  1. Why the authors just do association why can't PrePost changes be seen with RANOVA or so 

We reported pre- vs. postoperative changes of other sleep parameters (such as AHI and oxygen desaturation) in the earlier reports 1-4. The pre- vs. postoperative changes of arousal index were illustrated in Figure 7.

The aim of this study is to report how respiratory arousal presents in the distinct subgroup of very severe OSA, how it relates to AHI, ESS, BMI, and oxygen saturation, and how a non-framework surgery may change it. A one-way analysis of variance (ANOVA) on the percentage of postoperative change as a function of various sleep parameters may see if the percentage of postoperative changes are similar across sleep parameters. However, this is not part of this study. Please indicate specifically where we can improve.

Discussion 

  1. Discussion needs to be modified accordingly 

We revised the 2nd paragraph in Discussion as: “In addition to AHI, the results also show correlation of respiratory arousal index vs. each of mean SpO2, mean desaturation, and desaturation index; but not in BMI or ESS. In these patients, the respiratory arousal index explained 29.5%, 15.5%, 27.7%, and 30.5% of the variance in AHI, mean SpO2, mean desaturation, and mean desaturation index, respectively. The results of correlation are compatible…”

Reviewer 2 Report

 First, the physiological advantages and disadvantages of respiratory arousal are clearly described.

 Figure 1 shows that upper airway resistance syndrome is unlikely to be included in the population, which is convincing for this study.

 The fact that the respiratory arousal index is not associated with BMI and ESS suggests that it is less associated with obesity and drowsiness, but it would be better to add that consideration.

 Regarding the association between hypoxia or hypopnea and respiratory arousal, hypoxia and hypercapnia during sleep may affect the respiratory arousal index. As the authors have mentioned, this is a predictable result.

 Importantly, the effectiveness of Non-Framework Surgery in very severe SAS patients has shown a significant improvement in the respiratory arousal index compared preoperatively and postoperatively. This is very good, but I think it would be better to add a consideration about the comparison of the effect with CPAP treatment and the proper use of surgical treatment.

Line 193: Please add some discussion about the results that were not associated with BMI or ESS and respiratory arousal, except for the small sample size.

Line 205: If this study shows improvement in sleep depth as well as respiratory arousal after Non-Framework Surgery, please add a brief note.

Line 216: Please also describe the comparison of the effect with CPAP treatment, which is a general treatment, and the proper use of surgical treatment and CPAP treatment in severe OSAS. 

Author Response

Enclosed is our revised manuscript. We have addressed the questions. The response to each question is listed as follows. Please note the reference numbers of the responses are indexed at the end of this letter and differ from those in the manuscript.

First, the physiological advantages and disadvantages of respiratory arousal are clearly described.

 Figure 1 shows that upper airway resistance syndrome is unlikely to be included in the population, which is convincing for this study.

Thank you for your careful reading. To prevent distractions, we did not discuss UARS.

 The fact that the respiratory arousal index is not associated with BMI and ESS suggests that it is less associated with obesity and drowsiness, but it would be better to add that consideration.

To provide this consideration, we revised the 2nd paragraph of Discussion to “… e.g., on correlation with BMI). The result of the respiratory arousal index not being associated with BMI and ESS suggests that it is less associated with obesity and drowsiness. It requires future studies…”

 Regarding the association between hypoxia or hypopnea and respiratory arousal, hypoxia and hypercapnia during sleep may affect the respiratory arousal index. As the authors have mentioned, this is a predictable result.

 Importantly, the effectiveness of Non-Framework Surgery in very severe SAS patients has shown a significant improvement in the respiratory arousal index compared preoperatively and postoperatively. This is very good, but I think it would be better to add a consideration about the comparison of the effect with CPAP treatment and the proper use of surgical treatment.

We added another paragraph in Discussion: “The level of how CPAP therapy reduces arousal index varies 5-7. In Kim’s study 7, arousal index was reduced 87.6% from 57.1 to 7.1 events/hour in 34 patients with severe OSA (mean 63.4 ± 17.5 event/hour). Ferguson et al. reported a reduction rate of 23.9% from 28.9 to 22.0 events/hour in 27 patients with moderate to severe OSA (mean AHI 24.5 ± 8.8 events/hour) 5. Randerath et al. reported a reduction rate of 35.3% from 21.8 to 14.1 events/hour in 20 patients with mild to moderate OSA (mean AHI 17. 5 ± 7.7 events/hour) after using CPAP for 6 weeks 6. It is difficult to compare these CPAP results with the surgery in the present study because of the lack of controls, such as disease severity, patient selection, and home or sleep center monitoring. For patients with an acceptable compliance with CPAP but ask for the surgery, it requires a future study to investigate how CPAP therapy may improve arousal in these patients before the surgery. The results may help the decision making in the preoperative discussion.”

Line 193: Please add some discussion about the results that were not associated with BMI or ESS and respiratory arousal, except for the small sample size.

We added this in the 2nd paragraph of Discussion: “… The result of the respiratory arousal index not being associated with BMI and ESS suggests that it is less associated with obesity and drowsiness. It requires future studies…”

Line 205: If this study shows improvement in sleep depth as well as respiratory arousal after Non-Framework Surgery, please add a brief note.

To better focus on the research question of how respiratory arousal presents in the distinct subgroup of very severe OSA and how the surgery may change it, we wish to report the effect of the surgery on sleep depth and efficiency in another paper.

Line 216: Please also describe the comparison of the effect with CPAP treatment, which is a general treatment, and the proper use of surgical treatment and CPAP treatment in severe OSAS. 

We added another paragraph for this point. Please see the last paragraph of Discussion.

Round 2

Reviewer 1 Report

adequate changes have been made still it can be improved 

Reviewer 2 Report

The authors have revised it in response to my request. A specific description was also added to the limitation of this study.